

# Determination of PaO2/FiO2 after 24 h of invasive mechanical ventilation and ΔPaO2/FiO2 at 24 h as predictors of survival in patients diagnosed with ARDS due to COVID-19

Miguel Hueda-Zavaleta[1,2], Cesar Copaja-Corzo[1,3], Brayan Miranda-Chávez[1], Rodrigo Flores-Palacios[2], Jonathan Huanacuni-Ramos[2], Juan Mendoza-Laredo[1,2], Diana Minchón-Vizconde[1,4], Juan Carlos Gómez de la Torre[5] and Vicente A. Benites-Zapata[6]

[1] Facultad de Ciencias de la Salud, Universidad Privada de Tacna, Tacna, Perú
[2] Hospital III Daniel Alcides Carrión—EsSalud, Tacna, Perú
[3] Red Asistencial Ucayali EsSalud, Ucayali, Perú
[4] Hospital Hipólito Unanue de Tacna, Tacna, Perú
[5] Laboratorio Clínico Roe, Lima, Perú
[6] Unidad de Investigación para la Generación y Síntesis de Evidencias en Salud, Universidad San Ignacio de Loyola, Lima, Peru

Corresponding authors
Miguel Hueda-Zavaleta,
miguelhueda90@gmail.com,
Mighueda@virtual.upt.pe
Vicente A. Benites-Zapata,
vbeniteszapata@gmail.com,
vbenites@usil.edu.pe

## ABSTRACT

**Introduction**. Acute respiratory distress syndrome (ARDS) due to Coronavirus Disease 2019 (COVID-19) causes high mortality. The objective of this study is to determine whether the arterial pressure of oxygen/inspiratory fraction of oxygen (PaO2/FiO2) 24 h after invasive mechanical ventilation (IMV) and the difference between PaO2/FiO2 at 24 h after IMV and PaO2/FiO2 before admission to IMV (ΔPaO2/FiO2 24 h) are predictors of survival in patients with ARDS due to COVID-19.

**Methods**. A retrospective cohort study was conducted that included patients with ARDS due to COVID-19 in IMV admitted to the intensive care unit (ICU) of a hospital in southern Peru from April 2020 to April 2021. The ROC curves and the Youden index were used to establish the cut-off point for PaO2/FiO2 at 24 h of IMV and ΔPaO2/FiO2 at 24 h associated with mortality. The association with mortality was determined by Cox regression, calculating the crude (cHR) and adjusted (aHR) risk ratios, with their respective 95% confidence intervals (95% CI).

**Results**. Two hundred patients were analyzed. The average age was 54.29 years, 79% were men, and 25.5% ($n = 51$) died. The cut-off point calculated for PaO2/FiO2 24 h after IMV and ΔPaO2/FiO2 24 h was 222.5 and 109.5, respectively. Those participants with a value below the cut-off point of ΔPaO2/FiO2 24 h and PaO2/FiO2 24 h after IMV had higher mortality, aHR = 3.32 (CI 95% [1.82–6.07]) and aHR = 2.87 (CI 95% [1.48–5.57]) respectively.

**Conclusion**. PaO2/FiO2 24 h after IMV and ΔPaO2/FiO2 24 h in patients diagnosed with ARDS due to COVID-19 on IMV were associated with higher hospital mortality. These findings are helpful to identify those patients with a higher risk of dying on admission to the ICU.

# INTRODUCTION

A significant percentage of patients with Coronavirus Disease 2019 (COVID-19) develop severe disease with respiratory failure, and some of the progress to acute respiratory distress syndrome (ARDS), requiring admission to invasive mechanical ventilation (IMV) in an intensive care unit (ICU) (*World Health Organization (WHO), 2021*; *Copaja-Corzo et al., 2021*). The lethality in these patients can vary from 28.6% to 60.4% (*Grasselli et al., 2020*; *Petrilli et al., 2020*; *Schuijt et al., 2021a*).

ARDS is characterized by severe hypoxemia, reduced lung compliance, and bilateral pulmonary infiltrates on chest radiography (*Villar, 2011*). Because the diagnosis is based on different parameters, in 2012, the Berlin criteria were proposed (*Ranieri et al., 2012*), which use the relationship arterial oxygen partial pressure/inspired oxygen fraction (PaO2/FiO2) to classify the severity of the condition.

Currently, there is evidence that hypoxemia is associated with an increase in mortality in patients diagnosed with COVID-19; multiple studies have determined an association between a lower PaO2/FiO2 or oxygen pulse oximetry at hospital admission with an increase in-hospital mortality in patients with COVID-19, which can be up to 7 times higher than in patients without hypoxemia (*Hueda-Zavaleta et al., 2021*; *Mejía et al., 2020*; *Santus et al., 2020*). However, the variation in PaO2/FiO2 that would impact after 24 h of admission to IMV on hospital outcomes is still unknown. In this context, we aimed to know if the difference between PaO2/FiO2 at 24 h after IMV and PaO2/FiO2 before admission to IMV (ΔPaO2/FiO2 24 h) is associated with mortality. In addition, we evaluate whether the increase or decrease of the PAO2/FIO2 24 h after the VMI was associated with greater or lesser mortality in patients diagnosed with ARDS by COVID-19.

# MATERIALS AND METHODS

## Study design and site

A retrospective cohort study was conducted using physical and electronic medical records of patients diagnosed with ARDS due to COVID-19. The study was carried out at Hospital III Daniel Alcides Carrión, located in the district of Calana in Tacna, in southern Peru. This third-level referral hospital belongs to the Tacna Social Security Health Care Network (EsSalud). It has a capacity of 28 beds in intensive care units (ICU) (*EsSalud, 2022*). The study period was from April 2020 to April 2021. The writing of the manuscript was carried out following the recommendations of STROBE to inform observational studies (*Vandenbroucke et al., 2007*).

## Study population

The study population included critically ill adult patients (≥18 years) diagnosed with ARDS due to COVID-19 on IMV and hospitalized in the ICU. Diagnosis of COVID-19 was

made by real-time polymerase chain reaction (RT-PCR) or positive nasopharyngeal swab antigenic rapid test. The diagnosis of ARDS was made according to what was proposed in the Berlin criteria (*Ranieri et al., 2012*). The patients who could not confirm the diagnosis of COVID-19, who were still in the ICU during data collection, and in whom complete data were not available in their medical record records were excluded. The sample included all patients who met the eligibility criteria within the study period.

## Hospital management

As part of the institutional protocol for critical patient ventilatory management in our hospital, all patients admitted to IMV underwent initial programming with the following parameters in pressure-controlled mode: (a) tidal volume 6-8ml/kg of ideal weight, (b) Positive end-expiratory pressure (PEEP): 8–12 cmH2O, (c) Respiratory rate (RR) for Carbon dioxide arterial pressure (PaCO2) between 35 and 50 mmHg and or pH >7.2 and (d) FiO2 for pulse oximetry saturation (SatO2) between 92–96%. Additionally, all patients were placed in the prone position for 48 to 72 continuous hours, which was suspended when PaO2/FiO2 was ≥200 mmHg with PEEP ≤10 and FiO2 ≤40%. Likewise, all patients received sufficient analgesia to allow the coupling to the IMV, associated with neuromuscular blockade during the prone period. On the other hand, those patients who, after coming out of pronation, presented a PaO2/FiO2 <150 or a 20% decrease in PaO2/FiO2 with respect to PaO2/FiO2 in prone (with or without posterior dense pulmonary infiltrates evidenced in the multicut computerized tomography) were subjected again to another cycle of prone.

We decided to evaluate all the patients who underwent MV and were admitted to the ICU and met the inclusion criteria. We calculated the statistical power of the collected sample (200 patients) based on the study by *Schuijt et al. (2021a)*, who reported mortality at 28 days in patients with severe ARDS on the second day after admission to IMV of 44.3% and 28.6% in patients with mild ARDS, also reported a ratio of unexposed to exposed of 6.21 (764/123). With a confidence level of 95% and 200 participants in the study sample and a non-exposed/exposed ratio of 3, calculated the statistical power was at 53.5%.

## Data collection

After reviewing all physical and digital medical records, patients were evaluated from hospital admission to outcome (medical discharge or death). Sociodemographic, clinical, ventilatory (before access to the IMV and 24 h later), laboratory and imaging characteristics that the radiologists reported at the time of admission to the IMV were collected, the calculation of the extent of parenchymal involvement was performed by a semi-quantitative scoring used in previous studies (*Chang et al., 2005*; *Pan et al., 2020*). To calculate lung damage, the average percentage of lung damage reported for each lobe was used (from 0 to 100%). Complications and therapies administered during the ICU stay were also evaluated.

Two researchers conducted data collection; Each one entered the data independently; A third researcher supervised and compared the data collected; when he found any error, the third researcher reviewed the medical history and corrected the error. The coded data collection forms were entered into the Microsoft Excel v.2016 program and performed the

statistical analysis with the STATA V17.0 (StatCorp, College Station, TX, USA) and Prism 9 (GraphPad, San Diego, CA, USA) software.

## Analyzed variables
### Result variable

The outcome variable was hospital mortality, this data was collected from the medical records of the patients included in the study, and the data were compared with the national death registry of Peru.

### Exposure variables

*Clinical characteristics and treatment.* The clinical characteristics were sex, age, comorbidities (obesity, DM2, hypertension, heart failure, asthma, chronic kidney disease, and immunosuppression), the number of comorbidities and date you entered the emergency room (2020/2021). The treatment administered was antibiotics, corticosteroids, colchicine, tocilizumab, and renal replacement therapy; these data were obtained from digital medical records.

*Ventilatory and laboratory characteristics.* The ventilatory characteristics were obtained from the analysis of arterial gases that were brought before the patient entered mechanical ventilation; the rest of the parameters were obtained from the report given by the mechanical ventilator and collected by the ICU nursing staff.

The laboratory characteristics were obtained from the record given in the digital medical records.

*Complications during ICU stay.* The most important complication was ARDS, defined according to the Berlin criteria (*Ranieri et al., 2012*). The rest of the complications were: nosocomial infection, sepsis, septic shock, acute kidney injury, and arrhythmias. All were diagnosed according to international criteria (*Copaja-Corzo et al., 2021*; *Singer et al., 2016*; *Turagam et al., 2020*; *Khwaja, 2012*).

## Statistical analysis

Categorical variables were described as absolute and relative frequency, and quantitative variables as mean/standard deviation or median/interquartile range depending on whether their distribution was normal or not. The Chi-square or Fisher's exact statistical tests were used for the bivariate analysis between the categorical variables and the results. For the quantitative variables, the Student's $t$-test was used if the assumptions were fulfilled or, failing that, the Mann–Whitney U test. For the comparison with the degrees of severity of ARDS, the ANOVA or Kruskal Wallis statistical tests were used according to the normality of the variables.

We investigated the predictive capacity of $\Delta$PaO2/FiO2 24 h and PaO2/FiO2 24 h after IMV as a predictor of survival in patients with ARDS due to COVID-19. First, we calculated the Receiver Operating Characteristic (ROC) curves, the ROC curve (area under the curve, AUC) area, and the corresponding Youden indices. The highest Youden index identified the optimal cut-off point. Next, the 95% CI was calculated by bootstrapping a thousand repetitions, thereby generating dichotomous exposure variables for $\Delta$PaO2/FiO2 24 h

and PaO2/FiO2 24 h after IMV. Then, we calculated the sensitivity, specificity, and odds ratio with their respective 95% CI of the established cut-off points to predict survival. In addition, the differences between the ROC curves were tested using the Chi-square test (*Safari et al., 2016*).

Finally, to answer our hypothesis, we used Cox proportional hazards models to find the crude and adjusted hazard ratios (aHR) and their respective 95% confidence intervals (95% CI). The variables ΔPaO2/FiO2 24 h, PaO2/FiO2 24 h of IMV, and ARDS severity before IMV and after 24 h of IMV, as they were not independent of each other, were analyzed individually in different adjusted regressions (one for each variable). In the adjusted model, we included the variables: age, sex, number of comorbidities, septic shock, acute renal failure, positive end-expiratory pressure, plateau pressure, driving pressure, and date of admission. These variables were selected for clinical plausibility as they could influence the outcome. The proportionality of the multivariate model had a value of $p = 0.452$.

### Ethical aspects

This research followed the international guidelines of the Declaration of Helsinki. The research protocol was evaluated and approved by the ethics committee of the Faculty of Health Sciences of the Private University of Tacna (Identification code: N391-2021-UPT/FACSA-D). Due to the retrospective observational nature of the study, they did not request informed consent. All information was coded to maintain the anonymity of the participants, and only the researchers had access to the data.

## RESULTS

### Characteristics of the study population

Of the 235 medical records examined, 35 did not meet our inclusion criteria (25 were still hospitalized and 10 had incomplete data), so a total of 200 medical records of patients with ARDS secondary to COVID-19 undergoing IMV and who were admitted to the ICU; of these the 79% ($n = 158$) were male, and the mean age was 54.29 (SD: 12.19) years. A total of 78% ($n = 159$) of those admitted had at least one comorbidity, and the most frequent were obesity (59%), hypertension (26.5%), and diabetes mellitus type 2 (21.50%). The median length of hospital stays, time in the ICU, and time in IMV was 20 (IQR: 14–28) days, 10 (IQR: 5.5–16) days, and 10 (IQR: 6–16) days, respectively (Table 1).

At the time of hospital admission, the median SatO2 and PaO2/FiO2 were 88% (IQR: 84.5–90) and 245.5 (IQR: 165.5–297), respectively, and the mean extent of pulmonary involvement on chest tomography was 61.34% (SD: 14.02). At the time of admission to IMV, the median PaO2/FiO2 was 91.5 (IQR: 70–129.5); likewise, 58.5% of patients were admitted to IMV with a diagnosis of severe ARDS, and the median SOFA score at admission was 4 (IQR: 3–5) points. Regarding the laboratory data, the median of lymphocytes, procalcitonin, and lactate dehydrogenase were 675.65 (IQR: 480–1070.4) cells/mm3, 0.14 (IQR: 0.06–0.37) ng/mL, and 757.5 (IQR: 587.5–990.5) IU/L, respectively (Table 1).

At 24 h after admission to IMV, the median PaO2/FiO2 was 240 (IQR: 200–301.5), and the 24 h ΔPaO2/FiO2 was 141 (105–193). Concerning ventilatory parameters, the median
**Table 1  Demographic, clinical, ventilatory and laboratory characteristics of the study population and comparison between survivors and deceased.**

| Variable | Total (n = 200) | Survivors (n = 149) | Non-survivors (n = 51) | p-value |
|---|---|---|---|---|
| Demographic characteristics | | | | |
| Age (years)[*] | 54.29 (±12.19) | 52.14 (±11.91) | 60 (±10.89) | <0.001[a] |
| - <50 years (%) | 74 (37.00) | 63 (42.28) | 11 (21.57) | 0.019[b] |
| - 50–59 years (%) | 58 (29.00) | 42 (28.18) | 16 (31.37) | |
| - ≥60 years (%) | 68 (34.00) | 44 (29.54) | 24 (47.06) | |
| Sex | | | | 0.777[b] |
| - Female (%) | 42 (21.00) | 32 (21.48) | 10 (19.60) | |
| - Male (%) | 158 (79.00) | 117 (78.52) | 41 (89.40) | |
| Number of comorbidities[**] | 1 (1–2) | 1 (0–2) | 2 (1–3) | 0.001[c] |
| - No comorbidity (%) | 44 (22.00) | 40 (26.84) | 4 (7.85) | 0.004[b] |
| - Only 1 (%) | 79 (39.50) | 60 (40.27) | 19 (37.25) | |
| - Two or more (%) | 77 (38.50) | 49 (32.89) | 28 (54.90) | |
| Comorbidities | | | | |
| - Obesity (%) | 118 (59.00) | 88 (59.06) | 30 (62.74) | 0.976[b] |
| - DM2 (%) | 43 (21.50) | 27 (18.12) | 16 (31.37) | 0.047[b] |
| - AHT (%) | 53 (26.50) | 35 (23.49) | 18 (35.29) | 0.099[b] |
| - Heart failure (%) | 10 (5.00) | 2 (1.34) | 8 (15.68) | <0.001[d] |
| - Asthma (%) | 30 (15.00) | 17 (11.41) | 13 (25.49) | 0.015[b] |
| - Chronic kidney disease (%) | 8 (4.00) | 1 (0.67) | 7 (13.72) | <0.001[d] |
| - Inmunosuppressión (%) | 18 (9.00) | 7 (4.70) | 11 (21.57) | 0.001[b] |
| Date of admission | | | | <0.001[b] |
| - 2020 | 90 (45.00) | 52 (43.90) | 38 (74.51) | |
| - 2021 | 110 (55.00) | 97 (65.10) | 13 (25.49) | |
| Clinical and ventilatory characteristics | | | | |
| - Hospital stay time (days)[**] | 20 (14–28) | 20 (14–27) | 19 (14–29) | 0.721[c] |
| - Time from hospitalization to IMV (days)[**] | 2 (1–4) | 2 (1–4) | 2 (1–4) | 0.760[c] |
| - Time in ICU (days)[**] | 10 (5.5–16) | 9 (5–13) | 14 (10–26) | <0.001[c] |
| - Time in IMV (days)[**] | 10 (6–16) | 8 (5–12) | 15 (11–26) | <0.001[c] |
| - SatO2 (%)[**] | 88 (84.50–90) | 89 (85–91) | 85 (80–90) | 0.001[c] |
| - PaO2/FiO2 ratio at hospital admission[**] | 245.5 (165.50–297) | 254 (179–301) | 218 (128–262) | 0.008[c] |
| - PaO2/FiO2 before entering IMV | 91.50 (70–128) | 95 (77–130) | 68 (55–113) | 0.001[c] |
| - Shock and need for vasopressors (%) | 70 (35.00) | 34 (22.81) | 36 (70.59) | <0.001[b] |
| - PaO2/FiO2 24 h after IMV[**] | 240 (200–301.50) | 260 (222–310) | 190 (144–240) | <0.001[c] |
| - ΔPaO2/FiO2 24 h[**] | 141 (105–193) | 152.2 (116–198) | 106 (61–156) | <0.001[c] |
| - Tidal volume 24 h after IMV[**] | 460 (427.50–506.50) | 460 (425–500) | 474 (430–540) | 0.323[c] |
| - PEEP 24 h after IMV[**] | 12 (10.50–14) | 12 (10–13) | 13 (12–14) | 0.003[c] |
| - Inspiratory pressure 24 h after IMV[**] | 15 (14–18) | 15 (14–18) | 15 (14–18) | 0.551[c] |
| - Plateau pressure 24 h after IMV[**] | 28 (26–31) | 28 (26–30) | 30 (28–34) | <0.001[c] |
| - Driving pressure 24 h after IMV | 16 (15–18) | 16 (14–18) | 18 (15–21) | 0.005[c] |
| - Number of pronations | | | | 0.001[b] |

| Variable | Total (n = 200) | Survivors (n = 149) | Non-survivors (n = 51) | p-value |
|---|---|---|---|---|
| - One | 127 (63.82) | 104 (69.80) | 23 (45.10) | |
| - More tha one | 72 (36.18) | 44 (30.20) | 28 (54.90) | |
| - Time in pronation | | | | 0.148[d] |
| - ≤48 h | 182 (91.00) | 138 (92.62) | 44 (86.27) | |
| - >48 h | 18 (9.00) | 11 (7.38) | 7 (13.73) | |
| - Lung damage on CT (%)[*] | 61.34 (±14.02) | 58.12 (±12.89) | 70.76 (±12.98) | <0.001[a] |
| - SOFA upon admission to IMV | 4 (3–5) | 4 (3–4) | 5 (4–5) | <0.001[c] |
| - Severity of ARDS at 24 h of IMV | | | | <0.001[d] |
| - Mild (%) | 149 (74.50) | 126 (84.56) | 23 (45.10) | |
| - Moderate (%) | 45 (22.50) | 23 (15.44) | 22 (43.13) | |
| - Severe (%) | 6 (3.00) | 0 (0.00) | 6 (11.77) | |
| Laboratory characteristics at admission to IMV | | | | |
| - Leukocytes (cells/mm3)[**] | 9920 (6950–12630) | 9240 (6870–12000) | 10610 (7100–16170) | 0.111[c] |
| - Lymphocytes (cells/mm3)[**] | 675.65 (480–1070.40) | 680 (490–1062.80) | 663.2 (405.90–1078.20) | 0.890[c] |
| - Platelets (cells × 10³/L)[**] | 283 (217–354) | 293 (235–366) | 243 (199–305) | 0.001[c] |
| - CRP (mg/dL)[**] | 13 (6–17.40) | 12.43 (5.35–17.53) | 14.19 (10.31–17.07) | 0.130[c] |
| - Procalcitonin (ng/mL)[**] | 0.14 (0.06–0.37) | 0.12 (0.05–0.33) | 0.22 (0.12–0.50) | 0.019[c] |
| - LDH (U/L)[**] | 757.50 (587.50–990.50) | 727 (585–940) | 886 (642–1240) | 0.016[c] |
| - CPK-Total (U/L)[**] | 129 (66–260) | 120 (64–251) | 161 (79–300) | 0.238[c] |
| - CPK-MB (U/L)[**] | 26.25 (20–35) | 26 (19.95–34) | 29.2 (21.20–53) | 0.084[c] |
| - AST (U/L)[**] | 51.50 (31–82) | 50 (31–79) | 53 (30–84) | 0.997[c] |
| - ALT (U/L)[**] | 64 (39–120) | 74 (42–140) | 56 (34–79) | 0.004[c] |
| - Total bilirubin (mg/dL)[**] | 0.66 (0.46–0.92) | 0.64 (0.45–0.90) | 0.67 (0.50–1) | 0.362[c] |
| - Creatinine (mg/dL)[**] | 0.84 (0.7–1.01) | 0.82 (0.70–0.94) | 0.95 (0.80–1.18) | 0.001[c] |

**Notes.**

[*]Mean and standard deviation.
[**]Median and interquartile range.
[a]Student's *T* for equal variances.
[b]Chi2.
[c]*U*-Mann Whitney.
[d]Fisher's exact.

DM2, Type 2 diabetes mellitus; AHT, Arterial hypertension; ICU, Intensive care unit; IMV, Invasive mechanical ventilation; SatO2, Oxygen saturation; PaO2/FiO2, the ratio between the oxygen pressure over the inspired fraction of oxygen; SOFA, sequential organ failure assessment score; CRP, C-reactive protein; LDH, Lactate dehydrogenase; CPK, Creatinine phosphokinase; AST, Aspartate aminotransferase; ALT, Alanine aminotransferase.

values for PEEP, plateau pressure, and driving pressure were 12 (IQR: 10.5–14), 28 (IQR: 26–31), and 16 (IQR: 15–18), respectively. Likewise, 74.5% of patients admitted to IMV were classified as having mild ARDS at 24 h (Table 1).

During the ICU stay, 29% (n = 58) of the patients presented a nosocomial infection, with pneumonia associated with IMV being the most frequent (75.86%); however, 98.5% of the patients' received antibiotics. Furthermore, 14.5% of the patients developed acute renal failure, and 51% required renal replacement therapy. Finally, 51 patients (25.5%) of those admitted to IMV died (Table 2).

**Table 2  Complications and therapy received during the ICU stay in the study population and comparison between survivors and deceased.**

| Variable | Total (n = 200) | Survivors (n = 149) | Non-survivors (n = 51) | p-value |
|---|---|---|---|---|
| Complications during ICU stay | | | | |
| Nosocomial infection (%) | 58 (29.00) | 22 (14.76) | 36 (70.58) | <0.001[a] |
| Type of superinfection | | | | |
| - Pneumonia associated with MV (%) | 44 (22.00) | 14 (9.39) | 33 (64.70) | <0.001[a] |
| - Catheter-associated bacteremia (%) | 11 (5.50) | 6 (4.02) | 5 (9.80) | 0.152[b] |
| - UTI associated with catheter (%) | 3 (1.50) | 2 (1.34) | 1 (1.96) | 0.998[b] |
| Sepsis (%) | 143 (71.50) | 99 (66.44) | 44 (86.27) | 0.007[a] |
| Septic shock (%) | 58 (29.00) | 24 (16.10) | 34 (66.66) | <0.001[a] |
| Acute kidney failure (%) | 29 (14.50) | 8 (5.37) | 21 (41.17) | <0.001[a] |
| Arrhythmia (%) | 13 (6.50) | 4 (2.68) | 9 (17.64) | 0.001[b] |
| Therapies received during the ICU stay | | | | |
| - Antibiotics (%) | 197 (98.50) | 146 (97.98) | 51 (100.00) | 0.572[b] |
| - Corticosteroids (%) | 194 (97.49) | 144 (96.64) | 50 (98.03) | 0.998[b] |
| - Colchicine (%) | 52 (26.00) | 34 (22.81) | 18 (35.29) | 0.080[a] |
| - Tocilizumab (%) | 26 (13.00) | 14 (9.39) | 12 (23.52) | 0.010[a] |
| - Renal replacement therapy (%) | 15 (7.50) | 2 (1.34) | 13 (25.49) | <0.001[b] |

Notes.

*Median and interquartile range.

[a]Chi2.

[b]Fisher's exact.

ICU, Intensive Care Unit; MV, Mechanical Ventilation; UTI, Urinary Tract Infection.

## Bivariate analysis according to the severity of ARDS

Significant differences were observed concerning greater severity of ARDS after 24 h in IMV in elderly patients, a greater number of comorbidities. The most frequent in this group were heart failure, chronic kidney disease, and immunosuppression. The clinical and ventilatory characteristics associated with greater severity of ARDS were, entering the emergency room with minor oxygen saturation, requiring vasopressors, a higher percentage of lung damage, and a higher SOFA score (Table S1).

## Bivariate analysis according to death

In the bivariate analysis, statistically significant differences were observed concerning mortality in patients with older age, with the presence of 1 or more comorbidities, and among them, mainly type 2 diabetes mellitus, heart failure, and chronic kidney disease. It was also observed that those who died had a more extended stay in the ICU (median: 14 days *vs* nine days) and IMV (median: 15 days *vs* eight days), lower SatO2, lower PaO2/FiO2 at hospital admission, a greater pulmonary involvement in the chest tomography and a higher score in the SOFA score. Likewise, the patients who died presented a lower PaO2/FiO2 before entering IMV (median: 68 *vs* 95; $p < 0.001$) and in the same way, at 24 h after IMV, PaO2/FiO2 and $\Delta$PaO2/FiO2 24 h were lower in the deceased with a median of 190 *vs* 260 ($p < 0.001$) and a median of 106 *vs.* 152.2 ($p < 0.001$), respectively, than in the survivors (Fig. 1). At 24 h, the proportion of patients with mild ARDS was 84.56% in the survivors and 45% in the deceased ($p < 0.001$). (Table 1). It was also observed that deceased patients
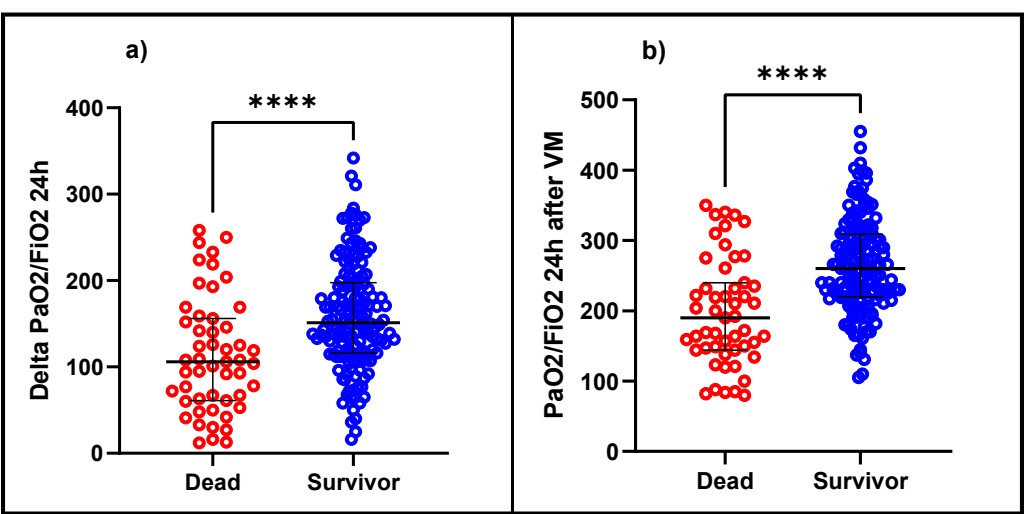

**Figure 1** **Difference between (A) ΔPaO2/FiO2 24 h and (B) PaO2/FiO2 24 h after IMV in patients diagnosed with ARDS due to COVID-19 between deceased and survivors.** PaO2/FiO2: fraction between the arterial pressure of oxygen over the inspired fraction of oxygen, ΔPaO2/FiO2 24 h: the difference between PaO2/FiO2 at 24 h after IMV and PaO2/FiO2 before admission to IMV.

presented higher plateau pressure, driving pressure, and higher PEEP values at 24 h of IMV and a higher incidence of ventilator-associated pneumonia, septic shock, arrhythmias, and acute renal failure (Table 2).

## Determination of predictive value of PaO2/FiO2 24 h after IMV and ΔPaO2/FiO2 24 h

PaO2/FiO2 24 h after IMV has an AUC ROC of 0.75 (95% CI [0.66–0.83]), with 222.5 (95% CI [175.85–269.14]) being the best cut-off point to determine survival, with a sensitivity and specificity of 74.5% (IC 95% 66.7–81.3) and 68.6% (IC 95% [54.1–80.9]), respectively. Similarly, the AUC ROC of ΔPaO2/FiO2 24 h was 0.70 (95% CI [0.61–0.78]), with a cut-off point of 109.5 (95% CI [85.43–133.56]), whose sensitivity and specificity were 81.2% (IC 95% [74–87.1]) and 56.9% (42.2–70.7), respectively. Although the AUC ROC of PaO2/FiO2 24 h after IMV was higher than ΔPaO2/FiO2 24 h, no statistically significant differences were observed between both curves ($p = 0.054$) (Fig. 2, Table 3).

## Multivariable model

In the crude Cox regression analysis, the variables associated with mortality were moderate ARDS (cHR: 2.68, 95% CI [1.49–4.83]) and severe ARDS (cHR: 13.98, 95% CI [5.59-34-93]) at 24 h of IMV, ΔPaO2/FiO2 24 h <109.5 (HRc: 3.70, CI 95% [2.12–6.46]), PaO2/FiO2 24 h after IMV <222.5 (HRc: 3.41, CI 95% [1.88–6.19]).

In the adjusted model, an aHR 3.32 (95% CI [1.87–6.07]) was obtained for ΔPaO2/FiO2 24 h <109.5, an aHR 2.87 (95% CI [1.48–5.57]) for PaO2/FiO2 24 h <222.5 after IMV and an aHR: 2.18 (CI 95% [1.13–4.12]) and aHR: 9.81 (CI 95% [3.43–28.04]) for moderate and severe ARDS after 24 h of IMV, respectively (Table 4).

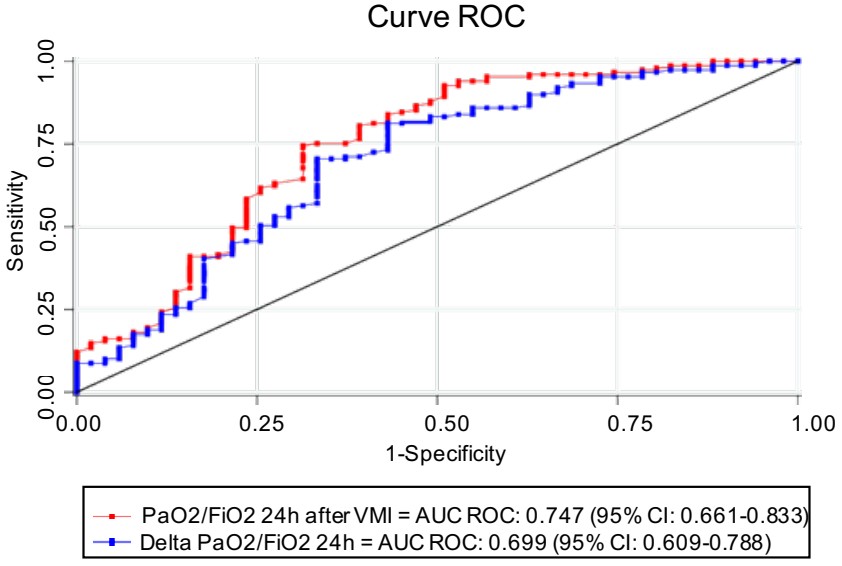

**Figure 2** **ROC curve and AUC ROC of PaO2/FiO2 24 h after IMV and ΔPaO2/FiO2 24 h to predict survival.** AUC, area under the curve, ROC, receptor operating characteristic, PaO2/FiO2, fraction of arterial oxygen pressure over inspired fraction of oxygen, ΔPaO2/FiO2 24 h: difference between PaO2/FiO2 at 24 h after IMV and PaO2/FiO2 before entering IMV.

**Table 3** **Cox regression analysis to assess predictors of death in patients hospitalized for critically ill COVID-19.**

| Variable | cHR (95% CI) | p-value | aHR (95% CI) | p-value |
|---|---|---|---|---|
| ΔPaO2/FiO2 24 h <109.5 | 3.70 (2.12–6.46) | <0.001 | 3.32 (1.82–6.07) | ≤ 0.001 |
| PaO2/FiO2 24 h IMV <222.5 | 3.41 (1.88–6.19) | <0.001 | 2.87 (1.48–5.57) | 0.002 |
| ARDS upon admission to IMV | | | | |
| Moderate | Ref | | Ref | |
| Severe | 1.40 (0.75–2.61) | 0.280 | 0.92 (0.47–1.79) | 0.806 |
| ARDS at 24 h of IMV | | | | |
| Mild | Ref | | Ref | |
| Moderate | 2.68 (1.49–4.83) | 0.001 | 2.18 (1.13–4.12) | 0.020 |
| Severe | 13.98 (5.59–34.93) | <0.001 | 9.81 (3.43–28.04) | <0.001 |

Notes.

cHR, crude Hazard ratio; aHR, adjusted Hazard ratio; IMV, invasive mechanical ventilation; PaO2/FiO2, fraction between arterial oxygen pressure over the inspired fraction of oxygen; ΔPaO2/FiO2 24 h, the difference between PaO2/FiO2 at 24 h after IMV and PaO2/FiO2 before admission to IMV; ARDS, Acute respiratory distress syndrome.
The variables ΔPaO2/FiO2 24 h, PaO2/FiO2 24 h of IMV, and severity of ARDS at admission to IMV and 24 h later, as they were not independent of each other, were analyzed individually. The variables were adjusted by: Age, sex, number of comorbidities, septic shock, acute renal failure, positive end-expiratory pressure, plateau pressure, driving pressure, and date of admission.

# DISCUSSION

In the present retrospective cohort study in patients diagnosed with ARDS and critical COVID-19 undergoing IMV and admitted to the ICU, overall hospital mortality of 25.5% was observed. Our findings show that those patients who responded with a significant increase in oxygenation and classified as having mild ARDS after 24 h of IMV had lower

**Table 4   Area under the ROC curve and cut-off points with the highest AUC ROC of the variables PaO2/FiO2 24 h after MV and ΔPaO2/FiO2 24 h to predict survival in patients hospitalized for critically ill COVID-19.**

|  | PaO2/FiO2 24 h after MV (IC 95%) | ΔPaO2/FiO2 24 h (IC 95%) |
|---|---|---|
| AUC ROC[*] | 0.75 (0.66–0.83) | 0.70 (0.61–0.78) |
| Cutpoint | 222.5 (175.85–269.14) | 109.5 (85.43–133.56) |
|   - Sensitivity at cutpoint % | 74.5 (66.7–81.3) | 81.2 (74.0–87.1) |
|   - Specificity at cutpoint % | 68.6 (54.1–80.9) | 56.9 (42.2–70.7) |
|   - AUC ROC at cutpoint | 0.72 (0.64–0.79) | 0.69 (0.65–76.58) |
|   - Odds ratio at cutpoint | 6.39 (3.20–12.75) | 5.70 (2.87–11.31) |
|   - Youden index | 0.431 | 0.381 |

**Notes.**
AUC, area under the curve; ROC, receptor operating characteristic; PaO2/FiO2, fraction of arterial oxygen pressure over the inspired fraction of oxygen; MV, Mechanical ventilation; ΔPaO2/FiO2 24 h, the difference between PaO2/FiO2 at 24 h after MV and PaO2/FiO2 before entering MV.
[*]$p = 0.054$.

hospital mortality than those with moderate ARDS and severe ARDS. Likewise, we identified the capacity of ΔPaO2/FiO2 24 h and PaO2/FiO2 24 h after IMV as predictors of survival in patients with ARDS and critical COVID-19.

In our study, the proportion of patients admitted to IMV with severe ARDS was approximately double that reported in other cohorts in France (*COVID-ICU Group on behalf of the REVA Network and the COVID-ICU Investigators, 2021*), the Netherlands (*Schuijt et al., 2021b*), the United Kingdom (*Richards-Belle et al., 2020*), and Spain (*Ferrando et al., 2020*). However, hospital mortality in our cohort was like that observed in these studies. Non-invasive ventilation and high-flow oxygen are used in many hospitals to reduce the number of patients who ultimately require admission to IMV; Unfortunately, in our hospital, we did not have these therapeutic measures, which could have influenced the high number of patients who were finally admitted to IMV.

We also observed that VMI entry with a diagnosis of severe ARDS was not a factor associated with statistically significant mortality in the adjusted model of our study. These findings could suggest that the severity of the ARDS would not be a determining forecast factor in mortality, especially compared to the seriousness of the ARDS 24 h after entering VMI. But more similar studies are necessary to support these results.

Those patients who suffered from severe ARDS after 24 h of IMV had a higher risk of death. These findings are like those reported by the PRoVENT study in patients with ARDS due to COVID-19, where they observed that patients with severe ARDS on day 2 had higher mortality than patients with mild or moderate ARDS (*Schuijt et al., 2021a*). Similar findings were observed in the pre-COVID-19 era in the LUNG SAFE study (*Madotto et al., 2018*). In this study, they reported that patients with severe ARDS at day 2 had a substantially higher risk of death than those with mild and moderate ARDS and "resolved ARDS," a term that refers to those patients who no longer met the definition of ARDS. The ARDS at 24 h of IMV occurred in up to 24% of those admitted to this study. Some authors propose that these patients could constitute false positives for ARDS (*Villar et al.,*

*2015*), who initially met the diagnostic criteria without presenting the pathophysiological processes seen in classic ARDS. It is unknown if this phenomenon also occurs in patients with COVID-19.

In our study, PaO2/FiO2 24 h after IMV and ΔPaO2/FiO2 24 h were independent factors associated with mortality, not observed with PaO2/FiO2 before intubation. This coincides with previous studies in the pre-COVID-19 era, where determined that oxygenation indices after 24 h of IMV are better predictors of results in patients with ARDS (*Huber et al., 2020*; *Lai et al., 2016*). However, other studies in patients with critical COVID-19 observed that a lower PaO2/FiO2 ratio on admission to the ICU was a factor associated with higher mortality (*Grimaldi et al., 2020*; *Nassar et al., 2021*), which disagrees with our findings. Although in none of these studies, it is clear what cut-off point was used for this conclusion, nor the moment of its determination with respect to intubation.

We also found that PaO2/FiO2 24 h after IMV has a higher AUC than ΔPaO2/FiO2 24 h in predicting survival, although at the expense of lower sensitivity. The AUC ROC of PaO2/FiO2 24 h after IMV observed in our study was higher than those described by other authors in the pre-COVID-19 era. *Lai et al. (2016)* reported that PaO2/FiO2 24 h after IMV presented an AUC ROC of 0.657 and was higher than that observed on day 0 of IMV (AUC ROC: 0.556). Similarly, *Huber et al. (2020)* reported that the AUC ROC of the Berlin definition at 24 h was 0.664 and 0.644 at 48 h. To our knowledge, there are no studies that have evaluated the AUC of PaO2/FiO2 at 24 h after IMV or ΔPaO2/FiO2 at 24 h in patients diagnosed with ARDS due to COVID-19 in IMV. An investigation in patients with ARDS due to COVID-19 in IMV evaluated the mortality prognostic capacity of pulse oximetry saturation/FiO2 (SatO2/FiO2) on days 1, 2, and 3 of IMV, whose AUC was 0.53, 0.62, and 0.62 respectively (*Roozeman et al., 2021*), which makes it inferior as a predictive metric than PaO2/FiO2 24 h after IMV and ΔPaO2/FiO2 24 h observed in our study. Likewise, we must highlight that all the patients in our study were placed in the prone position. It is likely that this intervention increased PaO2/FiO2 after IMV, as reported by previous observational studies in intubated patients with COVID-19 (*Shelhamer et al., 2021*; *Langer et al., 2021*) and should be considered in future research.

On the other hand, in our environment, as in most countries with limited resources, we do not have advanced strategies for the management of refractory hypoxemia such as extracorporeal circulation membrane (ECMO), high-frequency oscillatory ventilation (HFOV), or mechanical ventilators (*Ma et al., 2020*) with advanced modes of mechanical ventilation, which could be considered as an early alternative tool for patients with PaO2/FiO2 less than 222.5 at 24 h of IMV and/or with ΔPaO2/FiO2 less than 109.5, in order to guarantee MV protective and prevent hypoxemia.

This research presented some limitations, including the observational and retrospective nature of the study, which did not allow the exclusion of selection bias, nor did it allow the examination of other parameters (such as mechanical power, compliance, oxygenation index, more inflammatory markers or harmful habits). Another limitation was that the study was conducted in a single health institution with a limited number of patients. Therefore, these results could not be extrapolated to the general population. Finally, it is possible that the different measures of ventilatory therapy, such as changes in PEEP and

FiO2, may have influenced PaO2/FiO2 at 24 h of IMV; even so, it was a factor associated with mortality in the adjusted model. Therefore, prospective, multicenter studies and external validation of our findings in other populations are required.

## CONCLUSIONS

PaO2/FiO2 24 h after IMV and $\Delta$PaO2/FiO2 24 h in patients diagnosed with ARDS due to COVID-19 on IMV were factors associated with higher hospital mortality. Therefore, PaO2/FiO2, 24 h after IMV, and $\Delta$PaO2/FiO2 24 h are useful variables to identify those patients with a higher risk of dying in patients with ARDS due to COVID-19 in IMV.

### Funding

Universidad San Ignacio de Loyola financed the Article Processing Charge of the journal. The remainder of the study was self-funded. The funders had no role in study design, data collection and analysis, decision to publish, or preparation of the manuscript.

### Grant Disclosures

The following grant information was disclosed by the authors:
Universidad San Ignacio de Loyola financed the Article Processing Charge of the journal.

### Competing Interests

Juan Carlos Gómez de la Torre is a worker at the ROE clinical laboratory. The rest of the authors declare no conflict of interest.

### Author Contributions

- Miguel Hueda-Zavaleta conceived and designed the experiments, performed the experiments, prepared figures and/or tables, authored or reviewed drafts of the article, and approved the final draft.
- Cesar Copaja-Corzo conceived and designed the experiments, performed the experiments, analyzed the data, authored or reviewed drafts of the article, and approved the final draft.
- Brayan Miranda-Chávez performed the experiments, analyzed the data, authored or reviewed drafts of the article, and approved the final draft.
- Rodrigo Flores-Palacios performed the experiments, prepared figures and/or tables, and approved the final draft.
- Jonathan Huanacuni-Ramos performed the experiments, prepared figures and/or tables, and approved the final draft.
- Juan Mendoza-Laredo performed the experiments, prepared figures and/or tables, and approved the final draft.
- Diana Minchón-Vizconde performed the experiments, prepared figures and/or tables, and approved the final draft.

- Juan Carlos Gómez de la Torre performed the experiments, analyzed the data, prepared figures and/or tables, and approved the final draft.
- Vicente A. Benites-Zapata conceived and designed the experiments, analyzed the data, prepared figures and/or tables, authored or reviewed drafts of the article, and approved the final draft.

## Human Ethics

The following information was supplied relating to ethical approvals (*i.e.*, approving body and any reference numbers):

The institutional research ethics committee approved the Faculty of Health Sciences of the Private University of Tacna (Identification code: N391-2021-UPT/FACSA-D).

## Data Availability

The raw data is available in the Supplemental File.

## Supplemental Information

Supplemental information for this article can be found online at http://dx.doi.org/10.7717/peerj.14290#supplemental-information.

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
