# Peer review of "Determination of PaO2/FiO2 after 24 h of invasive mechanical ventilation and ΔPaO2/FiO2 at 24 h as predictors of survival in patients diagnosed with ARDS due to COVID-19"

_PeerJ, doi:10.7717/peerj.14290_

## Round 0.1 · original submission · Major Revisions

Authors should consider addressing the reviewers' concerns, mainly in relation to the experimental design.

·

Basic reporting

The manuscript is written in clear and professional English, it follows the usual structure including the layout of tables, figures and references.
Minor flaws: lines 156-158 are in Spanish; reference 15 is a duplicate of reference 5; in table 1, “HTA” is written where it should probably read “AHT”, and “VMI” where it should read “IMV”.
Raw data seem to be shared in a STATA database file. Since I don’t use STATA, I could not open and verify it.

Experimental design

The authors try to answer a research question with importance both clinical and scientifical – can survival with ARDS due to COVID-19 be predicted from oxygenation parameters early after the start of invasive mechanical ventilation? The basis for this study, its methods and conduction are clearly described, conclusions concerning the research question are derived from the results. However, I have some questions concerning the design and the hospital management that are not mentioned in the manuscript but might be crucial for the accuracy and relevance of the results.
Major issues:
If all patients were placed in the prone position for 48 or more hours: this study might actually show and analyze improvement of oxygenation due to proning rather than IMV. How many patients were still in the prone position at 24 h? Was proning used before intubation in some patients? Had additional proning in the later course of intensive care an effect on outcome? Were any clinical or radiographic parameters used to decide about later additional proning?
How long was the interval between hospital admission and intubation?
How were patients treated during this interval (e. g., high flow oxygen, non-invasive ventilation, proning, pharmaceuticals)?
How was the paO2/FiO2 ratio determined before intubation? Were arterial samples drawn from all patients at hospital admission? How was FiO2 determined in non-ventilated patients?
In what proportion of cases was the COVID-19 diagnosis made by antigen test instead of PCR?
How many patients were actually excluded because they were still in the ICU or because complete data were not available?
More other risk factors of mortality could and should have been mentioned and analyzed, at least as possible confounders, e. g., inflammatory markers, vital parameters (respiratory rate, temperature), smoking habit [e. g., Gozdas et al. J Int Care Med 2022 DOI: 10.1177/08850666221100411; Jakob et al. Infection 2022 DOI 10.1007/s15010-021-01656-z].
Was there a time effect? Intensive care for COVID-19 and its success changed during the pandemic, due to increasing experience, differences between virus subtypes, emergence of therapeutic modalities. The final months of the study might even have been influenced by vaccination.
Non-ventilatory COVID-19 therapy modalities described in table 2 are corticosteroids, colchicine and tocilizumab. Were antiviral drugs and neutralizing antibodies not given at all (I am unsure whether those were available at the time)?
I am not sure whether the regression analysis in table 3 should have been reported in greater detail since it carries the principal weight of the study’s results. I have not tried to reproduce this analysis.
Minor issues:
How was radiographic lung damage quantified? Was it really normally distributed to allow Student’s t-test?
Student’s t-test was also used for age although that is very likely not normally distributed.

Validity of the findings

No further comment (see "2. Experimental design").

·

Basic reporting

Thank you for the invitation. This manuscript attempts to establish the relationship between arterial
oxygen partial pressure/inspired oxygen fraction (PaO2/FiO2) and survival/mortality. I have a few comments on this paper

1. Authors have not provided a hypothesis that there is a need to ascertain the association of PaO2/FiO2 with survival or mortality. Is there any hypothetical evidence that variations in PaO2/FiO2 can relate to higher mortality? There is a need to strengthen the relationship of this study.

Experimental design

The authors have provided the details on the estimation of the statistical power of this study. But I did not find any calculations relating to the sample size estimation. Can the authors provide further details that how the sample size and power of this were estimated?
Since COVID-19 with ARDS is the target population of this study, the sample size of 200 seems low.
I would suggest authors please include a heading of operational definitions for this manuscript. Under this heading, authors can define the terms used in this study i.e. COVID-19, ARDS, mortality, etc...
Is this study a retrospective cohort or retrospective cross-sectional?
Please provide the details of the study site in the method section.
I will suggest authors include the following headings in the methods, ethics, study design and site, study population with inclusion and exclusion criteria, data collection (how data was collected, information on data collection forms), management of data, and missing variables, operational definitions, statistics.
The information on the bivariate analysis is not present in the statistical section (how the predictive variables were selected - either on clinical plausibility, reported in previous literature, or random testing). Which variables were selected for multivariate model?.

Validity of the findings

Table 1: The comorbid conditions were classified as no, 1 condition, or more than 2 conditions. I have a question here where those patients who had 2 conditions were classified?
are authors sure that for SOFA upon admission to VMI the p values were estimated by chi-square or Fisher exact test?

I am unable to read the results as given "Al momento de la admisión hospitalaria, la mediana de SatO2 y PaO2/FiO2 fueron de 88%
157 (RIC: 84.5-90) y 245.5 (RIC: 165.5-297) respectivamente, y la extensión media del
158 compromiso pulmonar en la tomografía de tórax fue de 61.34% (SD: 14.02)."

PaO2/FiO2 24 hours after IMV and PaO2/FiO2 24h were found as predictors of mortality. However, the authors did not discuss how these high-risk patients can be managed and how the hazards of mortality can be reduced in the group of patients.

Additional comments

The small sample size is a potential limitation of this study which is not discussed at the end of the discussion section. I have seen some variables having a frequency of 5, 4, or 3 which may render the power of analysis quite weak. The authors need to discuss these limitations and should provide suggestions for future research.

---

## Round 0.2 · Major Revisions

The authors have made an important and successful effort to improve their manuscript. However, there are still some issues raised by reviewer 1 which had not been fully addressed and that deserve further attention.

·

Basic reporting

Was already good in the first version, improved in the revised one.
Minor (typographical) flaws:
Table 1 “Características demográficas” should be translated into English. Time from hospitalization to IMV for non-survivors is given as 1 [median] (2-4) [inter-quartile range] which is clearly impossible.
Tables 1 and 2 Percentages are calculated linewise, so what they tell is, e.g., of all patients who were less than 50 years old, 85% were in the survivors’ group and 15% in the non-survivors’. With the two groups unequal in size, I would find it easier to read if percentages were calculated groupwise, so that it would tell, e.g., in the survivors’ group, 42 % were less than 50 years old, compared to 22% in the non-survivors’.

Experimental design

Response to the authors' response:
9. How about the timing of the drug therapy? Did some patients receive those drugs earlier than others?
13. It may not be the most crucial information, but I find it hard to accept that in a study “conducted using physical and electronic medical records” it should be impossible to determine which of two quite differently reliable methods was used to reach the principal diagnosis of this study. In the methods section, lack of confirmation for the diagnosis of COVID-19 is reported as an exclusion criterion. So, if it could not be confirmed at all during data collection for the study …?
15. Again, I find it disturbing what kind of parameters should be inavailable. Even if, in an emergency, no one thought of asking about a patient’s smoking habit, respiratory rate and body temperature should clearly be available from a medical record!
16. Even if vaccination is out of the picture for this study, intensive care of COVID-ARDS and its success changed over time. So, the individual date of admission (or start of IMV) should be included in the multifactorial analysis as a possible confounder.
19. I have heard of quantitative analyses of lung CT in COVID-19, but they are not yet established in clinical routine. Certain software add-ons can be utilized to quantify lung damage, and even to distinguish between ground-glass opacity and consolidation. To understand what the single percentage value in Table 1 means, the reader needs to know which method or software your radiologists employed.
Age is not normally distributed in the general population. Higher age is a known risk factor for severe or fatal COVID-19. In my opinion, age should not be regarded as normally distributed for this study.

Validity of the findings

No additional comment (see 2. Experimental design).

·

Basic reporting

The authors have addressed all my concerns.

Experimental design

The authors have addressed all my concerns.

Validity of the findings

The authors have addressed all my concerns.

Additional comments

The authors have addressed all my concerns.

---

## Round 0.3 · Minor Revisions

The manuscript has been greatly improved, however there are still a few minor issues raised by reviewer 1 to be addressed.

·

Basic reporting

Overall, clear and professional English, well-structured manuscript, helpful tables and figures.

There are some linguistic flaws, most of which don’t seem serious to me, e.g.:
Line 69: “In addition, we evaluated [how?] the PaO2/FiO2 24 hours after IMV impacts mortality…”
Line 153: “…test Student’s t-statistics were used…”

However, I stumbled over some sentences that I could only try to understand from their context:
Line 121: “The clinical history was reviewed again when placed and corrected the error.”
Legend of table 3: “The variables adjusted each one:…”

Since I am not a native English speaker myself, I may not have found all such semantic problems.

May I humbly suggest that a final version of this manuscript be copyedited by someone proficient in English?

Experimental design

For the quantitative analysis of CT scans, a semi-quantitative scoring is cited from previous studies. Said scoring delivers a value ranging between 0 and 25 that cannot be easily transformed into percentages which are reported in this study.

Validity of the findings

Percentages in table 1 were corrected as suggested, except for the number and times of pronations.

Since non-invasive ventilation and high-flow oxygen are routinely used in COVID-19 in many hospitals, it should be made known that they were never used in the patients studied here. That might have contributed to the higher rate of admission to IMV compared to other studies.

Line 265: I suggest a more cautious conclusion: discarding ARDS severity at IMV onset as a prognostic factor altogether may be premature and not supported by these small numbers. However, this study indicates that ARDS severity after 24 hours of IMV may be a better predictor.

Additional comments

You regarded some of my points of critique and improved your analysis and presentation accordingly. Other drawbacks I had pointed out could obviously not be helped because you could or would not access additional data from patient files that had been left out during original data collection.

---

## Round 0.4 · accepted · Accept

The authors have exhaustively addressed the remaining concerns raised by reviewer 1.

·

Basic reporting

See below and my previous reviews

Experimental design

See below and my previous reviews

Validity of the findings

See below and my previous reviews

Additional comments

Every change that could be made without repeating data acquisition has now been made. The remaining typographical and similar minor flaws are now a case for copyediting.